# Sequential fear generalization and network connectivity in trauma exposed humans with and without psychopathology

Xi Zhu[1,2,11], Benjamin Suarez-Jimenez[3,11], Amit Lazarov [1,4], Sara Such[5], Caroline Marohasy[3], Scott S. Small[1,2,6], Tor D. Wager[7], Martin A. Lindquist[8], Shmuel Lissek[9] & Yuval Neria [1,2,10]✉

While impaired fear generalization is known to underlie a wide range of psychopathology, the extent to which exposure to trauma by itself results in deficient fear generalization and its neural abnormalities is yet to be studied. Similarly, the neural function of intact fear generalization in people who endured trauma and did not develop significant psychopathology is yet to be characterized. Here, we utilize a generalization fMRI task, and a network connectivity approach to clarify putative behavioral and neural markers of trauma and resilience. The generalization task enables longitudinal assessments of threat discrimination learning. Trauma-exposed participants (TE; $N = 62$), compared to healthy controls (HC; $N = 26$), show lower activity reduction in salience network (SN) and right executive control network (RECN) across the two sequential generalization stages, and worse discrimination learning in SN measured by linear deviation scores (LDS). Comparison of resilient, trauma-exposed healthy control participants (TEHC; $N = 31$), trauma exposed individuals presenting with psychopathology (TEPG; $N = 31$), and HC, reveals a resilience signature of network connectivity differences in the RECN during generalization learning measured by LDS. These findings may indicate a trauma exposure phenotype that has the potential to advance the development of innovative treatments by targeting and engaging specific neural dysfunction among trauma-exposed individuals, across different psychopathologies.

[1] Department of Psychiatry, Columbia University Irving Medical Center, New York, NY, USA. [2] New York State Psychiatric Institute, New York, NY, USA. [3] Department of Neuroscience, University of Rochester, Rochester, NY, USA. [4] School School of Psychological Sciences, Tel-Aviv University, Tel-Aviv, Israel. [5] Department of Psychology, Pennsylvania State University, State College, PA, USA. [6] Department of Neurology, Columbia University Irving Medical Center, New York, USA. [7] Neuroscience Department, Dartmouth College, Hanover, NH, USA. [8] Department of Biostatistics, Johns Hopkins University, Baltimore, MD, USA. [9] Department of Psychology, University of Minnesota, Minneapolis, MN, USA. [10] Department of Epidemiology, Columbia University Irving Medical Center, New York, NY, USA. [11]These authors contributed equally: Xi Zhu, Benjamin Suarez-Jimenez. ✉email: ny126@cumc.columbia.edu

More than one third of people exposed to traumatic events, such as wars, disasters, and assaults, are likely to develop significant psychopathology, including post-traumatic stress disorder (PTSD), panic disorder (PD), generalized anxiety disorder (GAD), and major depression disorder (MDD)[1–4]. As symptoms of these disorders frequently overlap[5], their biological underpinnings are often shared[6,7]. Addressing the question of whether trauma exposure is associated with a distinct neural signature across psychopathologies such as those previously listed may advance our understanding of the corresponding neural dysfunction. Yet, efforts to advance knowledge regarding trauma-related neural aberrations have been hampered by adhering to traditional diagnostic systems, such as the Diagnostic and Statistical Manuals of Mental Disorders (DSM)[8], neglecting more objective markers including those clarified by brain imaging[9]. Furthermore, extant research examining neural biomarkers of trauma exposure has focused almost solely on patients with PTSD, limiting our understanding of potential shared mechanisms across psychopathologies versus trauma-exposed healthy controls. As such, this focus on PTSD has also complicated efforts to clarify whether resilience (i.e., exposure to trauma without developing significant psychiatric symptoms) may have a unique neural signature, with research on resilience being relatively sparse[10–12]. In an effort to address these gaps in knowledge, the present study aimed to clarify the behavioral and network functional magnetic resonance imaging (fMRI) markers of trauma and resilience in trauma-exposed people with and without psychopathology, and among people who were never exposed to trauma.

Previous studies have implicated excessive threat generalization (termed also threat or conditioned overgeneralization) as a potential endophenotype of several psychopathologies, including PTSD, PD, and GAD[13,14]. Threat generalization and discrimination are essential for learning about future threats. Generalization transfers a conditioned response from one stimulus to another similar, yet different, stimulus, such as concentric circles[15]. In threat-related fear responses, individuals who overgeneralize transfer fear from a dangerous stimulus to a less dangerous or even safe one, reflecting excessive fear and resulting in threat discrimination difficulties[15]. A recent meta-analysis study has shown significantly heightened behavioral fear generalization across trauma-, stress-, and anxiety-related disorder participants including GAD, PTSD, PD, SAD than controls, however, this study was limited to behavioral and psychophysiological data only[16].

Several brain areas have been identified to be involved in distinct computational process that are responsible for threat generalization and discrimination. These include the hippocampus[17,18], ventral prefrontal cortex (vPFC)[1,13], insula[19], dorsomedial prefrontal cortex (dmPFC)[20], anterior cingulate cortex (ACC), and thalamus[19]. Functional deficits in these brain areas, related to generalization/discrimination, have been attributed to PTSD[21,22]. The involvement of these neural areas also underscores the potential role of several essential brain networks in generalization/discrimination, including the default mode network (DMN), salience network (SN), and executive control network (ECN). These brain networks were also shown to play a key role in memory (DMN)[23], bottom-up (SN) and top-down executive control processes (ECN)[24].

Understanding the ways by which trauma exposure may alter these brain networks, and where they diverge from those of resilient individuals, is essential for clarifying the underlying basis of human psychopathology in the aftermath of trauma. To thoroughly elucidate behavioral and neural markers of trauma exposure and resilience, we utilized a generalization/discrimination task previously tested in patients with PD[14], GAD[25]

and PTSD[21]. These studies show that psychiatric patients, compared to normal controls, exhibit stronger generalization, implicating it as a putative marker of disrupted threat discrimination[13–15,23,25–27]. However, no study to date has assessed this neural marker among people exposed to trauma with psychiatric illnesses or among resilient participants who were exposed to trauma, but did not develop any significant psychiatric symptoms. In addition, no study to date has looked at how trauma exposed individuals learn to discriminate between cues over time. Learning to discriminate is an essential process whereby exposure to cues creates different neural representations for each cue. Understanding how these representations are built over time can give us an insight into how psychopathology develops and is maintained and how resilient trauma-exposed counterparts overcome it. Hence, the task used here entails two consequent generalization stages, termed early and late generalization (EG, LG), enabling to delineate the sequential trajectory of threat discrimination learning over these two stages. One study compared the trial-by-trial course of risk rating during generalization phase between PTSD and subthreshold PTSD to trauma-exposed controls[28]. This study provided initial evidence that relative to trauma-exposed controls, those with PTSD and subthreshold PTSD displayed significantly elevated generalization in early but not late trials[28]. However, research has yet to characterize the temporal associations between threat discrimination and network activation over the course of two consecutive generalization phases.

In the present study, aiming to characterize the behavior and neural markers of threat generalization/discrimination of trauma exposure and resilience, we assessed both the behavioral risk rating and network connectivity of the DMN, SN, and ECN, which has been largely unexplored in trauma-exposed populations. To clarify the behavioral and neural markers of trauma, trauma-exposed participants (TE) and healthy controls (HC) were compared. To identify the behavioral and neural markers of resilience, trauma-exposed participants with psychopathology (TEPG) were compared to trauma-exposed healthy controls (TEHC) and healthy controls (HC) with no trauma exposure. For behavioral markers, we hypothesized that there would be elevated generalization to stimuli resembling the CS+ in the TEPG group, compared to HC or TEHC groups. For neural markers, we expected that in the TEPG group there would be limited changes of network connectivity in SN and ECN over the two generalization stages, compared to TEHC or HC group.

## Results

**Demographics and clinical characteristics of the participants.** Demographic and clinical characteristics, including diagnoses, are presented in Table 1. As expected, compared to HC, TE participants had significantly higher PTSD symptoms[29], depression symptoms[30], SAD symptoms[31], panic disorder symptoms[32], GAD symptoms[33], and functional impairment[34]. In addition, TEPG, compared with TEHC, had higher psychopathology symptoms on all measures (all $p$'s < 0.001; See Table 1).

**Behavioral risk rating (conditioning).** First, we examined the appraisal of shock risk (risk ratings to CS+ vs. vCS−) in TE and HC separately, across the three phases (pre-acquisition, acquisition, and generalization), using paired sample t-tests. Results showed no CS+ vs. vCS- differences at the pre-acquisition phase in both groups (HC: $p = 0.09$, $t = 1.75$, df = 24; TE: $p = 0.39$, $t = -0.88$, df = 58). Conversely, a significant difference between the cues at acquisition (HC: $p = 0.009$, $t = -2.84$, df = 25; TE: $p < 0.001$, $t = -8.01$, df = 60) and generalization phase (HC: $p = 0.001$, $t = -3.83$, df = 25; TE: $p < 0.001$, $t = -9.95$, df = 60)

**Table 1 Demographic information of the sample.**

|  |  | HC | TE | TEPG | TEHC |
|---|---|---|---|---|---|
| N |  | 26 | 62 | 31 | 31 |
| Sex, N | Male | 15 | 37 | 19 | 18 |
|  | Female | 11 | 25 | 12 | 13 |
| Race, N | Black/African American | 12 | 29 | 13 | 16 |
|  | White/Caucasian | 10 | 15 | 9 | 6 |
|  | Asian | 1 | 6 | 2 | 4 |
|  | Others | 2 | 12 | 7 | 5 |
| Age, mean years (SD) |  | 34.58 (11.56) | 37.05 (12.35) | 37.19 (13.06) | 36.9 (11.81) |
| Education, mean (SD) |  | 15.83 (2.64) | 14.65 (2.47) | 14.36 (2.78) | 14.93 (2.14) |
| CAPS, mean (SD) |  | - | 17.74 (16.36) | 31.03 (11.95) | 4.45 (6.03) |
| HAM-D, mean (SD) |  | 0.4 (0.82) | 7.16 (7.26) | 12.43 (6.78) | 2.06 (2.56) |
| LSAS, mean (SD) |  | 11.54 (13.29) | 40.05 (35.06) | 59.55 (34.11) | 19.90 (22.66) |
| PDSS, mean (SD) |  | 0.46 (1.84) | 5.2 (6.16) | 8.97 (6.18) | 1.30 (2.84) |
| SF36, mean (SD) |  | 87.85 (7.63) | 66.62 (23.39) | 51.23 (20.28) | 82.53 (13.84) |
| GAD-7, mean (SD) |  | 1.08 (2.43) | 6.15 (6.16) | 9.94 (5.76) | 2.23 (3.60) |
| Current | PTSD | 0 | 26 | 26 | 0 |
|  | MDD | 0 | 11 | 11 | 0 |
|  | PDD | 0 | 5 | 5 | 0 |
|  | SAD | 0 | 1 | 1 | 0 |
|  | PD | 0 | 3 | 3 | 0 |
|  | OCD | 0 | 1 | 1 | 0 |
|  | ADHD | 0 | 1 | 1 | 0 |
|  | Insomnia | 0 | 1 | 1 | 0 |
|  | Eating Disorder | 0 | 1 | 1 | 0 |
|  | Subthreshold PTSD | 0 | 2 | 2 | 0 |
|  | Subthreshold MDD | 0 | 2 | 2 | 0 |

*CAPS* Clinician-Administered PTSD Scale-5, *HAM-D* Hamilton Depression Scale, *LSAS* Liebowitz Social Anxiety Scale, *PDSS* Panic Disorder Severity Scale, *PTSD* Posttraumatic Stress Disorder, *MDD* Major Depression Disorder, *PDD* Persistent Depressive Disorder, *SAD* Social Anxiety Disorder, *PD* Panic Disorder, *OCD* Obsessive-Compulsive Disorder, *ADHD* Attention Deficit Hyperactivity Disorder, *HC* Healthy Control, *TE* Trauma Exposed, *TEPG* Trauma Exposed Psychopathology Group, *TEHC* Trauma Exposed Healthy Control.

emerged in both groups (Figure S1, Table S2). These results suggest that both groups learned the CS contingencies.

### Threat overgeneralization/discrimination over time

*Trauma exposure markers*
Behavioral markers changes of stages: To examine the behavioral differences over the two generalization phases (EG, LG), we first assessed the changes of behavioral risk rating over the two stages (delta: EG-LG) of trauma exposure by comparing TE and HC using a group (TE and HC) by stimulus-type (vCS-, oCS-, GS1, GS2, GS3 and unreinforced CS+) repeated measures ANOVAs. Results indicated no significant findings of the changes of behavioral risk rating over the two stages ($p > 0.05$; Fig. 1).

Next, we assessed the changes of LDS across two stages using one-way ANOVA. Results indicated no significant findings ($p > 0.05$).

Neural markers changes of stages: To examine neural activity changes (delta: EG-LG) over the two generalization phases, we performed a group (TE, HC) by stimuli-type two-way ANOVA. No significant group by stimuli interaction was found. Only a corrected significant *main effect of group* emerged in SN and RECN (SN: $F = 7.01$, $p = 0.008$; RECN: $F = 10.25$, $p = 0.001$). This main effect of group was driven by HC displaying a higher reduction of SN and RECN activity, compared to TE who remained consistently higher SN and RECN activity, over the two generalization phases (Fig. 2, Fig. S2). Other networks indicated no significant findings.

Next, we accessed the changes of LDS across two stages in each of these ICNs using one-way ANOVA. Results indicated a significant group difference in SN LDS ($F = 7.492$, $p = 0.04$ Bonferroni corrected). This was driven by higher LDS reduction in HC compared to TE ($p = 0.04$ Bonferroni corrected; Fig. 3, Fig. S4).

Taken together, these analyses suggest that both groups were able to differentiate between the CS+ and the CS−, and rated the other generalization stimulus-type comparably during the task. However, compared to the TE, HC displayed a higher reduction of SN and RECN activity over the generalization stages, indicating a better learning effects. Additionally, compared with TE, HC displayed better discrimination learning over the two stages across different stimuli measured by LDS in SN. These findings suggest that while trauma exposure maintains high SN and RECN activity over time, it reduces the ability to use the SN to discriminate between the stimuli, as per the LDS.

*Resilience markers*
Behavioral markers changes of stages: To examine the behavioral differences over the two generalization phases (EG, LG), we first assessed the changes of behavioral risk rating over the two stages (EG-LG) of resilience by comparing TEHC, TEPG and HC using a group (TEHC, TEPG and HC) by stimulus-type (vCS−, oCS−, GS1, GS2, GS3 and unreinforced CS+) repeated measures ANOVAs. Results indicated no significant group by stimuli interaction of the changes of behavioral risk rating over the two stages (EG-LG; $p > 0.05$). A significant main effects group was found ($F = 6.69$, $p = 0.001$), where TEPG, compared to the other two groups, showed consistently higher risk ratings across the two stages (TEPG vs. HC: $p = 0.045$; TEPG vs. TEHC: $p = 0.005$ Bonferroni corrected; Fig. 1).

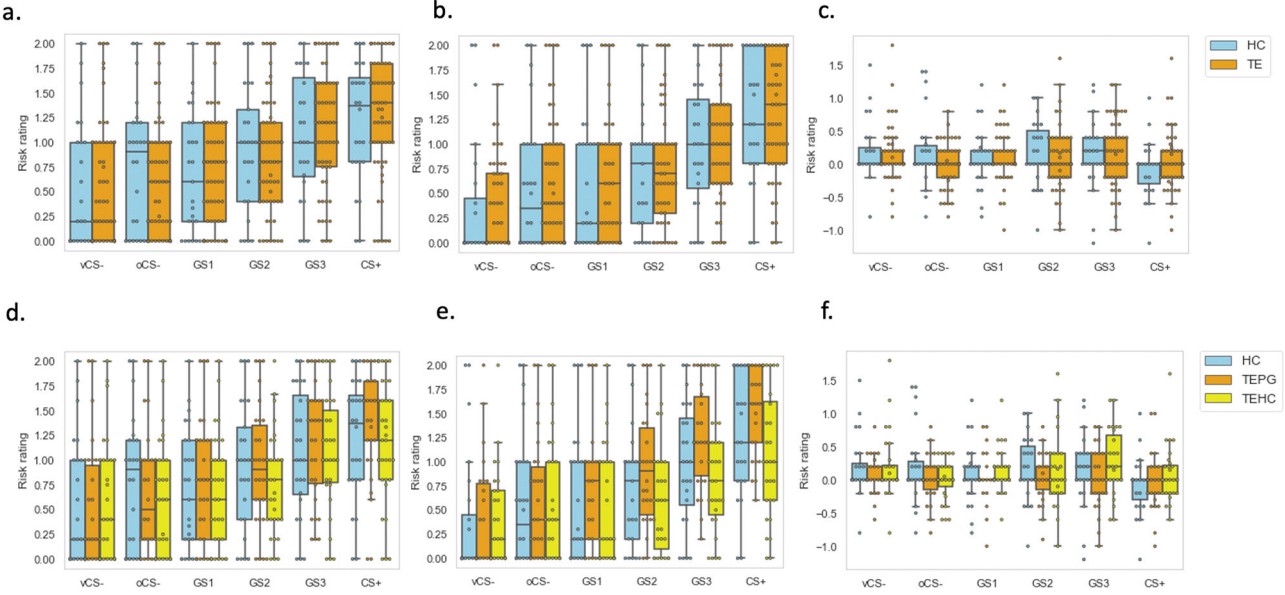

**Fig. 1 Behavioral risk rating of the generalization task. a** Risk rating in early generalization phase for TE and HC groups. **b** Risk rating in late generalization phase in TE and HC groups. **c** Risk rating of delta changes over time (EG-LG) in TE and HC groups. $n = 88$, error bar stand for ±2 standard error (SE). **d** Risk rating in early generalization phase in TEPG, TEHC and HC groups. **e** Risk rating in late generalization phase in TEPG, TEHC and HC groups. **f** Risk rating of delta changes over time (EG-LG) in TEPG, TEHC and HC groups. $n = 88$, error bar stand for ±2 standard error (SE). TE trauma-exposed participants, HC non-trauma-exposed healthy controls, TEHC trauma-exposed healthy controls, TEPG trauma-exposed psychopathology group, EG early generalization, LG late generalization.

Next, we assessed the changes of LDS using one-way ANOVA. Results indicated no significant changes of LDS, indicating no group discrimination learning differences during the generalization phase.

Neural markers changes of stages: To examine neural activity changes (delta: EG-LG) over the two generalization phases, we performed a group (TEPG, TEHC, HC) by stimuli-type two-way ANOVA. No significant group by stimuli interaction was found ($p > 0.05$). Only a corrected significant main effect of group emerged in SN and RECN (SN: $F = 7.13$, $p = 0.005$; RECN: $F = 12.74$, $p = 0.00002$ Bonferroni corrected). This main effect of group was driven by TEHC maintain higher SN and RECN activity, compared to TEPG (SN $p = 0.035$ Bonferroni corrected; RECN $p = 0.0006$ Bonferroni corrected) and HC (SN $p = 0.001$ Bonferroni corrected; RECN $p = 0.000015$ Bonferroni corrected) who displayed higher reduction of SN and RECN activity, over the two generalization phases (Fig. 2, Fig. S3). Other networks indicated no significant findings.

Next, we accessed the changes of LDS across two stages in each of these ICNs using one-way ANOVA. Results indicated a trending group difference in SN LDS ($F = 3.70$, $p = 0.029$) and RECN LDS ($F = 4.04$, $p = 0.021$). These results were driven by higher SN LDS reduction in HC group compared to the other two groups (HC vs.TEHC: $p = 0.018$, HC vs.TEPG: $p = 0.021$) and lower RECN LDS reduction in TEPG group compared to the other two groups (TEPG vs.HC: $p = 0.029$, TEPG vs. TEHC: $p = 0.01$; Fig. 3).

Taken together, these analyses suggest that while all groups could differentiate between the CS+ and CS−, TEPG, compared to TEHC and HC, had higher risk ratings particularly during the LG stage. Resilience findings could be explained by maintaining high SN and RECN activity over the two stages in the TEHC, compared to the TEPG and HC. Interestingly, higher LDS reduction of SN was found in the HC, compared to TEPG and TEHC, while higher LDS reduction was found in RECN in both the HC and TEHC groups, compared to TEPGs. That is, while

TEHC had maintained higher SN and RECN activity over time, the results show higher LDS in the RECN, suggesting a compensatory input used for discrimination. Overall, these results suggest that while trauma-exposed resilient individuals show a higher SN activity, a refined engagement of the RECN could potentially enhance their ability to successfully discriminate between stimulus-type resembling the one predicting threat.

## Discussion

To examine whether exposure to trauma is associated with measurable behavioral and neural markers this study utilized an fMRI generalization/discrimination task among trauma-exposed adults with and without psychopathology, as well as healthy none trauma-exposed controls. This study is the first to assess both behavioral and neural network markers of longitudinal threat overgeneralization and discrimination. Group differences between TE and HC revealed a distinct signature of trauma exposure. TEs, compared to HC, showed lower activity reduction in SN and RECN across two stage, and worse discrimination learning in SN measured by LDS. Additionally, group differences between TEPG, TEHC, and HC further revealed a resilience signature of network connectivity differences in the RECN during generalization learning. Overall, trauma-exposed resilient individuals demonstrated better discrimination learning over time in the RECN, compared with those who developed psychopathology. These findings may suggest a trauma exposure phenotype that has the potential to significantly advance treatment development by targeting specific well-delineated neural dysfunction among trauma-exposed individuals, across different psychopathologies.

When exploring the temporal sequence of generalization, HC and TEHC participants showed a good discrimination between GS and CS+, while TEPG exhibited higher risk ratings between GS3 and CS+ during generalization. Our behavior results suggest that the TEs showed overall higher risk ratings than HC. These results were driven by lower risk rating in TEHC and HC, compared to TEPG, further supporting a potential resilience

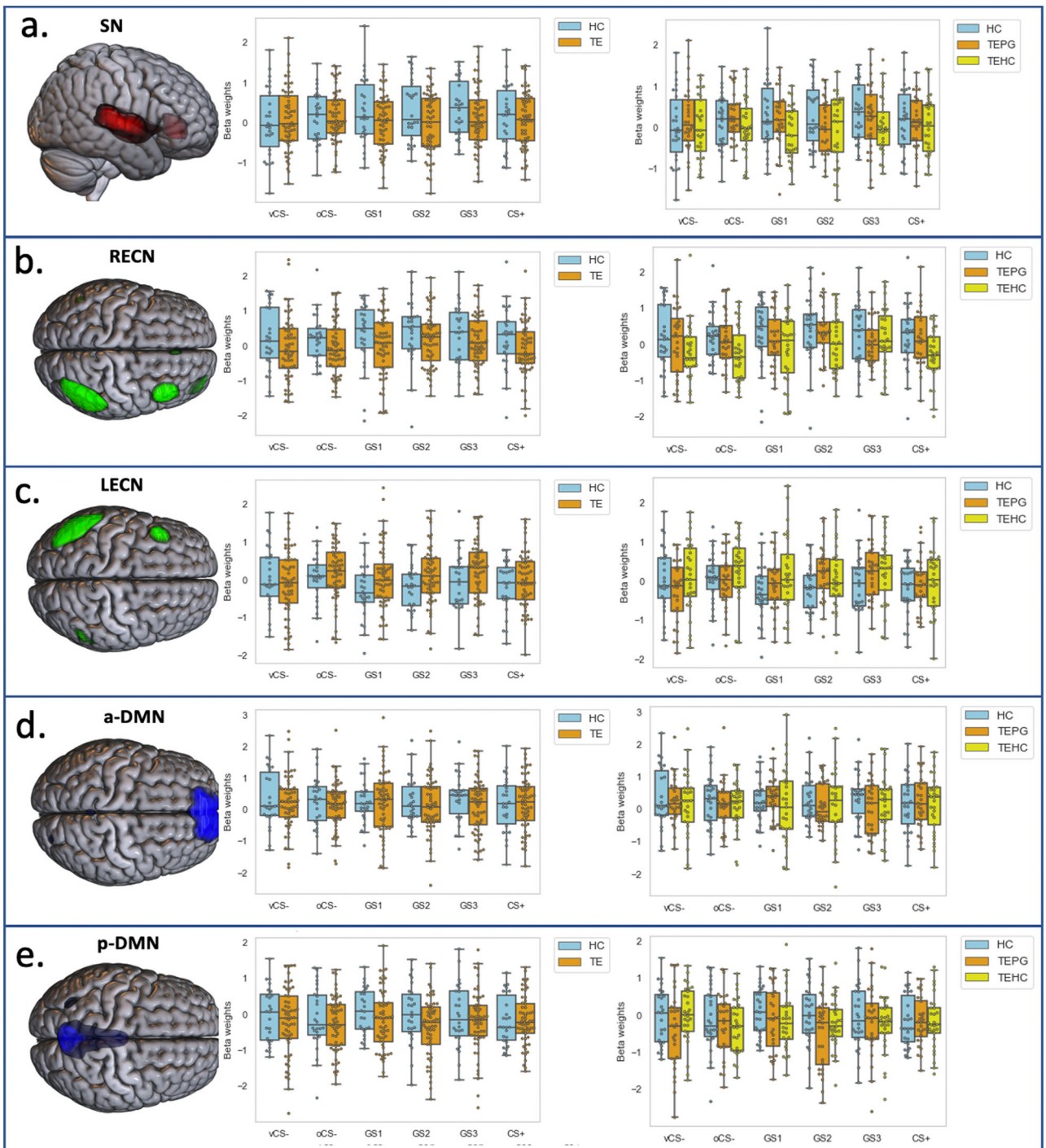

**Fig. 2 Neural activity changes (delta: EG-LG) over the two generalization phases.** Neural activity changes (delta: EG-LG) over the two generalization phases to conditioned and generalization stimuli (vCS-, oCS-, GS1, GS2, GS3, CS+) for (**a**) salience network (SN); (**b**) right executive control network (RECN); (**c**) left executive control network (LECN); (**d**) anterior default mode network (a-DMN); and (**e**) posterior default mode network (p-DMN) across two stages, $n = 88$, error bar stand for ±2 standard error (SE). CS+ conditioned stimuli, danger cue, oCS- conditioned stimuli, o shape safe cue, vCS-conditioned stimuli, v shape safe cue, GS generalization stimuli, HC non-trauma-exposed healthy controls, TEHC trauma-exposed healthy controls, TEPG trauma-exposed psychopathology group.

signature. The results support the premise that TEPG have a harder time discriminating between cues (overgeneralizing cues) particularly as they resemble the CS+. In turn, resilient trauma-exposed counterparts were able to discriminate between the cues comparably to HC. Indeed, our findings support previously reported discrimination deficits in PTSD[21] and anxiety[25], where PTSD groups show increased generalization to stimuli resembling

the CS+. These results suggest that trauma exposure by itself does not necessarily hamper one's ability to discriminate between cues, unless one develops psychopathology following exposure to trauma.

Studying generalization across the two stages, TE participants maintained higher overall activity in the SN, indicating a general trauma exposure marker. The SN plays an important role in

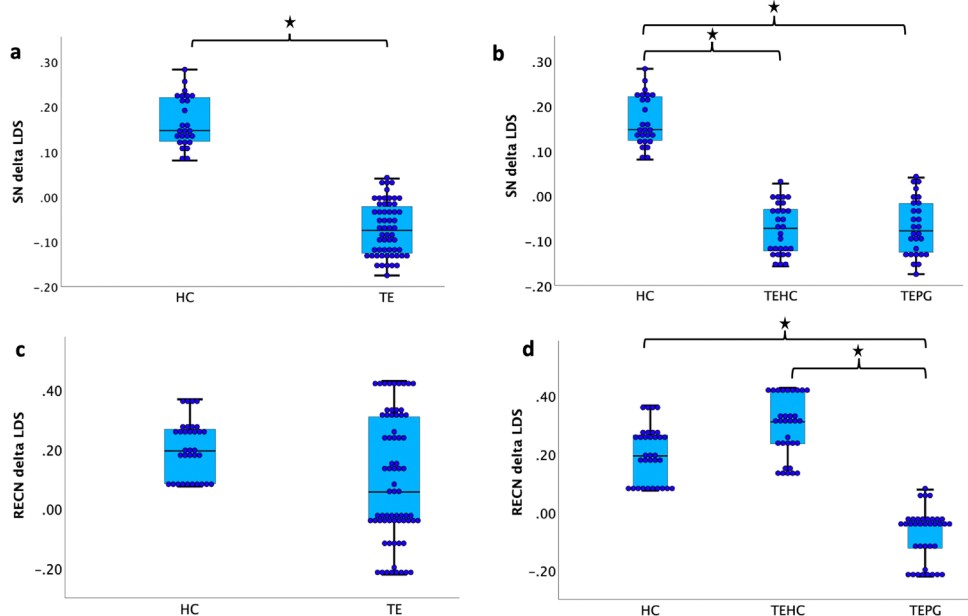

**Fig. 3 Neural activity changes measured by Linear deviation scores (LDS). a** LDS changes for SN in TE and HC groups. **b** LDS changes for SN in TEPG, TEHC and HC groups. **c** LDS changes for RECN in TE and HC groups. **d** LDS changes for RECN in TEPG, TEHC and HC groups. n = 83, error bar stand for ±2 standard error (SE). LDS linear deviation scores, SN salience network RECN right executive control network, HC non-trauma-exposed healthy controls, TEHC trauma-exposed healthy controls, TEPG trauma-exposed psychopathology group.

graded levels of threat-related salience detection[35,36], with usually highest levels to CS+ which gradually diminishes with decreasing CS+ resemblance. In this case, results show that HC have a higher reduction in SN activity over time, not evident in TE participants. Additionally, the HC group had higher LDS in the SN when compared to the TE group, further indicating a trauma exposure signature. Specifically, within the HC group, the identified neural signature of overgeneralization measured by steepness of gradients (LDS) was improved over time, suggesting better discrimination in the late generalization stage, compared with the earlier one. The SN in the TE group had less of a generalization gradient and remained the same over time, suggesting poor discrimination learning in the TE group. Interestingly, when TE participants were further divided into TEPG and TEHC, we found that this SN effect was driven by the TEHC group as compared to the other two groups (TEPG; HC). This is further supported by the SN LDS analysis, which revealed that in TEHC, like in TEPG, showed less discrimination learning measured by the change of LDS over the two stages. Overall, these results suggest that while HC are able to use the SN to improve the detection and discrimination of threat, trauma-exposed individuals cannot.

Examining the ECN, we found that TE maintained RECN activity from EG to LG. However in TEHC, like in HC, RECN was associated with increased discrimination learning (LDS). Generalization effects in the ECN may represent graded decreases in cognitive load that are proportional to decreases in fear reactivity as stimulus-type differentiate from CS+. According to attentional control theory, cognitive correlates of anxiety, including worry, attentional bias toward threat, and top-down efforts to manage anxiety, consume limited working-memory resources[37]. Our TEHC group findings, that the low RECN activity and the increased discrimination learning, suggest that less cognitive control network is needed as anxiety reduces over time. That is, trauma-exposed resilient individuals might rely on a refined detection and monitoring system to discriminate between stimulus-type, reducing their cognitive load, and therefore reducing their anxiety. The opposite is seen in the TEPG group, which

showed high SN and RECN activity and low discrimination learning associated with these networks. This may further suggests that the ECN is needed in order to regulate anxiety and fear to discriminate those stimulus-type which more resemble the CS-.

Several limitations of the present study should be noted. First, while we were expecting to find limbic area differences, particularly in the hippocampus, this was not the case. It could be that as the present task did not involve context, there was a lack of modulatory response by the hippocampus[38]. Future studies could emphasize context learning and discrimination following trauma exposure to test this hypothesis. Second, the data was collected in two scanners, and while we ensured that data was properly harmonized and controlled for scanner across all analysis, we cannot exclude the possibility that noise might be introduced when combining the two datasets. Third, we excluded patients with psychosis, schizophrenia, and bipolar illness for which there is some evidence for trauma history, which may limit the generalizability of obtained results. Future research may consider using more lenient inclusion/exclusion criteria by expanding the range of included disorders. Finally, and most interesting, it is tempting to consider the current results as potential targets for prevention or identifying risk-factors for the development of psychopathology after trauma-exposure. However, due to the limitations of cross-sectional studies, such as this one, we cannot infer whether the observed results in this study are caused by trauma-exposure or are developed prior to trauma-exposure. Future longitudinal studies should focus on these neural signatures, to identify their potential applicability for prevention of future psychopathologies.

In summary, our study shows evidence for both a trauma exposure signature and a resilience signature, not limited to specific psychiatric psychopathologies. The plasticity of threat and memory networks identified here may present significant opportunities for developing interventions aiming to target trauma exposure-related neural abnormalities. The SN and ECN activity emerged as key circuits of threat generalization following trauma exposure. However, higher activity in the SN in concert with higher RECN related to discrimination learning may represent a resilience signature in healthy trauma-exposed

individuals, who do not develop psychiatric disorders following a traumatic experience. Interventions directed at neurogenesis of the SN and ECN circuits might help increase discrimination in those who developed psychiatric disorders. Emerging research suggests that brain stimulation, particularly transcranial direct current stimulation (tDCS), a non-invasive technique, which elicited its action via multifaceted mechanisms, such as immunomodulation and neurogenesis, is a promising treatment for fear-related abnormalities by modulating threat memories and enhancing cognitive control via application to the prefrontal cortex[39]. Interventions, such as tDCS, of the SN and ECN may have relevance in reducing long term effects of trauma exposure and the development of post-trauma psychopathology.

## Methods

**Participants**. One hundred fourteen participants (79 TE and 35 HC) completed the study protocol (see Table 1 and Fig. S5). Of the 79 TE participants, 41 had psychopathology at the level of a psychiatric diagnosis (TEPG), and 38 were trauma-exposed healthy controls with no diagnosis (TEHC; see Table 1). Participants were recruited by local advertisements, websites, and word-of-mouth referrals and evaluated at the New York State Psychiatric Institute (NYSPI). The methods were performed in accordance with relevant guidelines and regulations and approved by NYSPI IRB. All participants provided written consent to take part in the study.

To ensure substantial range of psychopathology in the TE sample, and consistent with a large body of work suggesting dose-response relationships between trauma exposure severity and psychopathology[40], we methodically characterized type, severity, number and timing of trauma exposures, and included both childhood and adulthood trauma exposure. Index trauma exposure met DSM-5 Criterion A of a traumatic event.

An independent MD or PhD/PsyD clinical evaluator administered the Structured Clinical Interview for DSM-5 Disorders (SCID-5)[41] and the Clinician-Administered PTSD Scale-5 (CAPS-5)[29,42]. The CAPS-5 is a 30-item instrument containing 0-4 Likert-style items of frequency and intensity for PTSD symptoms. It has good reliability and validity[43]. Depressive symptoms were assessed using the Hamilton Depression Scale (HAM-D, 17 item)[30], a reliable clinician-rated measure of depressive severity. Social anxiety symptoms were assessed using the Liebowitz Social Anxiety Scale (LSAS), a well-validated 24-item clinician-rated scale for Social Anxiety Disorder (SAD) severity, widely used in studies of SAD[31]. Panic disorder symptoms were assessed using the Panic Disorder Severity Scale (PDSS)[32], a self-report measure for panic severity. Self-rated instruments also included the Generalized Anxiety Disorder assessment (GAD-7)[33], and the SF-36[34], a 36 item measure of generic health status, designed for use in clinical practice and research[44]. Out of the 62 TE participants, 26 had PTSD, 11 had MDD, 5 had persistent depressive disorder (PDD), 3 had PD and 1 had SAD (See Table 1).

Trauma-exposed participants were excluded due to: 1) Prior or current diagnosis of schizophrenia, psychotic disorder, bipolar disorder, dementia; (2) HAM-D score >25 reflecting significant depression and/or depression related impairment that warranted pharmacotherapy or combined medication and psychotherapy; (3) Individuals at risk for suicide; (4) History of substance/alcohol dependence within the past six months, or abuse within past two months; (5) Any psychotropic medications including antipsychotic, antidepressant, mood stabilizer, or stimulant medications in the last four weeks prior to the study (6 weeks for fluoxetine); (6) Pregnancy; (7) Medical illness that could interfere with assessment of response or biological measures (fMRI); (8) Paramagnetic metallic implants or devices contraindicating magnetic resonance imaging or any other non-removable paramagnetic metal in the body; and (9) Significant claustrophobia that would preclude ability to remain calm within the MRI scanner.

Healthy control participants were excluded due to: (1) Any current or past psychiatric diagnosis; (2) A history of trauma exposure that fulfills DSM-5 PTSD criteria A; (3) HAM-D score> 7; (4) Lifetime history of substance/alcohol dependence or abuse history; and point 6-9 as above.

**fMRI Task**. This task has been described in previous study by Lissek et al., 2014[23]. Briefly, this task consists of six types of stimuli including conditioned stimuli (CS+: danger cue; oCS−: o shape safe cue; vCS-: v shape safe cue) and three generalization stimuli (GSs including GS1, GS2, GS3). The GSs (i.e., GS1, GS2 and GS3) formed a continuum-of-size between the CS+and oCS− with GS3, GS2 and GS1 demarcating the GS with most to least similarity to the CS+, regardless of CS+ size. The inclusion of the vCS− allows for an assessment of brain responses to the CS+ (vs vCS−) that are independent of generalization effects to all ringed stimulus-type. Such an assessment is important because brain activations to the CS+ was used as functional regions of interest in which to test gradients of threat generalization and should thus be orthogonal to the generalization process. The dimensions and size increments for employed rings are described in Fig. S6. There were a string of colored crosshairs (blue, yellow, red, green, and purple) presented serially for a duration of 800 ms each, in a quasi-random order in the

center of the viewing screen during 4 s presentation of each stimulus. The inter-trial interval (ITI) periods last 2.4 s (three crosshairs) or 4.8 s (four crosshairs), during which participants focused their gaze on crosshairs in the center of the screen.

**Procedure**. Participants were not instructed of the CS/US contingency but were told they might learn to predict the shock if they attend to the presented stimulus-type. All CSs and GSs were presented for 4 s on a rear-projection viewing screen mounted at the foot of the scanner with a viewing distance of 6.71 feet (204.47 cm). The unconditioned stimulus (US) was a 100 ms electric shock (3–5 mA) delivered to the participant's right ankle. Prior to the start of the experiment, a sample shock procedure was performed during which participants received between one and three sample shocks and a level of shock rated by participants as being 'highly uncomfortable but not painful' was established. Shock intensity varied by participant and had an average intensity of 4.6 mA (s.d. = 0.80).

Next, participants practiced using the button box to respond to the red crosshairs, which appeared both at the center of CSs and GSs and during the ITI periods. Participants were then placed in the scanner. Participants were instructed to continuously monitor the stream of colored crosshairs and rate their perceived level of risk for shock as quickly as possible following each red cross using a three-button fiber optic response pad (Lumina LP-404 by Cedrus, San Pedro, Los Angeles, USA), where 0 = 'no risk', 1 = 'moderate risk' and 2 = 'high risk'. Risk ratings were recorded with Presentation software (Neurobehavioral Systems, Berkeley, CA, USA). Structural scans were acquired followed by pre-acquisition, acquisition, and generalization fMRI scans. Participants rated their anxiety to CS+, oCS− and vCS− after pre-acquisition, acquisition, and generalization between scans.

**Behavioral ratings**. Finally, self-reported anxiety to CS+, oCS−, and vCS− were retrospectively assessed following the pre-acquisition, acquisition, and generalization phases using a 10-point scale. The behavioral risk rating and neural network analysis excluded non-learner participants, indicated by rating the vCS- higher than CS+ during task-related risk rating or the post-task questionnaire[27].

**Design**. The generalization paradigm included three phases (Fig. S6): (i) pre-acquisition-consisting of 20 trials of each stimulus type (CS+, GS1, GS2, GS3, oCS-and vCS-), all presented in the absence of any shock US; (ii) acquisition-including 15 CS+, 15 oCS−, and 15 vCS−, with 12 of 15 CS+ co-terminating with shock (80% reinforcement schedule;[14] and (iii) generalization-including an early generalization (EG) stage and late generalization (LG) stage, each comprised of 10 trials of each stimulus type (unreinforced CS+, GS1, GS2, GS3, oCS−, vCS−) and an additional 5 CS+ co-terminating with shock (33% reinforcement schedule) to prevent extinction of the conditioned response while leaving 10 unreinforced CS+ to index responses uninfluenced by the shock US. Trials of all three phases were arranged in quasi-random order such that no more than two stimulus-type of the same class occurred consecutively. An additional constraint for the generalization phase was the arrangement of trials into six blocks of 13 trials each (i.e., two unreinforced CS+, one reinforced CS+, two oCS-, two vCS-, two GS1, two GS2, two GS3) to ensure an even distribution of trial types throughout runs.

**Image Acquisition and analysis**. Seventy-seven participants were scanned using a 3 T General Electric MR750, and 37 participants were scanned using a 3 T General Electric PREMIER (GE Medical Systems, Waukesha, WI, USA) equipped with a 32-channel receive-only head coil. For each participant a high-resolution T1-weighted 3D BRAVO sequence was acquired using the following parameter: T1 = 450 mm, Flip angle = 12°, field of view = 25.6 cm, 256 × 256 matrix, slice thickness = 1 mm. Whole brain T2*-weighted echo-planar images (EPIs) depicting the blood-oxygen-level-dependent (BOLD) were acquired for each participant with TR = 1.3 sec, TE = 28 msec, FA = 60°, FOV = 19.2 cm, number of slices = 27, slice thickness = 4 mm. A head cushion was used to limit head motion.

**Image Preprocessing**. All fMRI images were preprocessed using MATLAB version R2018a (The MathWorks, Inc., Natick, Massachusetts) and statistical parametric mapping software (SPM12; Welcome Trust Centre for Neuroimaging, UCL, London, United Kingdom). (1) Functional images were spatially realigned to the first image in the time series using a six-parameter rigid body transformation; (2) slice-time correction was performed; (3) outlier detection was carried out using artifact removal tools (ART). The principal component-based noise-correction method, "CompCor," implemented in this toolbox, was used for additional control of physiological noise and head motion effects. Outlier volumes in each participant were identified as having large spiking artifacts (i.e., volumes >3 standard deviations from the mean image intensity), or large motion (i.e., 0.5 mm for scan-to-scan head-motion composite changes in the x, y, or z direction); (4) each functional image was then spatially normalized to the standard T1 template included in SPM12; functional images were then resliced to 2 × 2 × 2 mm voxels, according to the resulting spatial realignment and normalization parameters; (5) anatomical images were segmented into grey matter, white matter, and cerebrospinal fluid (CSF) regions; (6) functional scans were smoothed with an 8 mm

full-width-at-half-maximum (FWHM) Gaussian kernel; (7) covariates corresponding to head motion (6 realignment parameters and their derivatives), outliers (one covariate per outlier consisting of 0 s everywhere and a 1 for the outlier time point), and the BOLD time series from the participant-specific white matter and CSF masks were used in the GLM and connectivity analysis as predictors of no interest, and were removed from the BOLD functional time series using linear regression.

From the original 114 sample, four participants were excluded due to being late for scan and not able to finish the generalization task (two participants), or task failure of delivering shock (two participants). Twenty-two additional participants (15 TE and 7 HC) were excluded with greater risk rating to vCS−, compared to CS+ during acquisition phase in the fMRI task, or greater rating to vCS−, compared to red cross in post questionnaires. No participant was excluded from further analysis because of movement exceeding ±1 mm. Consequently, the final neural imaging analysis included 88 participants: 62 TE and 26 HC. Sum of root mean square (RMS) of 6 relative head motion parameters (movement from this time point to the next one) was calculated for each participant in all groups (TEPG, TEHC, and HC). No significant difference in head motion was found between each pair of groups ($p > 0.5$). Linear regression was performed to study the linear relationship between the dependent variable (sum of head motion) and independent variable (groups). The regression analysis results showed that the total head motion could not predict groups ($p > 0.05$).

**fMRI network analysis**. Group independent components analysis was performed using the GIFT toolbox (v3.0b, http://icatb.sourceforge.net), implemented as a MATLAB toolbox (Matlab 2020b, MathWorks Inc., Sherborn, MA, USA), to obtain functional networks that underlie fMRI data. Two data reduction steps were performed using principal component analysis (PCA). First, participant-specific PCA was performed to reduce the dimensionality of each participant's functional data[45]. Second, participant data were concatenated into one group and PCA was performed again prior to performing ICA. Independent components, or networks, were calculated using the Infomax algorithm[46]. Number of independent components were estimated from the fMRI data by using the minimum description length (MDL) criteria, yielding 30 components. The infomax algorithm was repeated fifty times with randomly initialized decomposition matrices and the same convergence threshold using the ICASSO approach to assess the reliability of the generated components[47]. For each IC the"centroid" (i.e., the most stable result) was determined following the agglomerative hierarchical clustering with average-linkage criterion, and its consistency was calculated with a cluster quality index ($Iq$) ranging from 0 to 1[47]. Single-participant component time courses were then back reconstructed using the GICA-3 back-reconstruction approach.

A systematic approach was used to identify non-artifactual ICs, or intrinsic connectivity networks (ICNs)[48]. First, the $Iq$ index from ICASSO was assessed as the criterion to validate the IC decomposition stability. Components with an $Iq$ value less than 0.7 from 50 ICASSO repetitions were excluded. Second, visual evaluation of IC spatial patterns (e.g., ringing) as well as frequency inspection of IC time course spectra (e.g., time courses vastly dominated by low-frequency fluctuations) allowed additional components related to artifacts to be excluded from analysis. All the ICs that involved the majority of activation falling outside the cerebral cortex, for instance in the spinal cord, eyes, borders of the skull, ventricles, etc., were considered noise components and excluded from further analyses. After careful visual inspection of the spatial–temporal characteristics of each IC, 6 components were categorized as noise components, leaving 24 components for further analyses. Identification of the remaining components was accomplished by performing spatial correlation with publicly available GIFT network templates[49]. Each IC was correlated with the given templates and best network template was selected based on the maximum correlation values. To identify the components most involved in each trial type, a GLM analysis was performed. We examined the role of each ICN for each condition (CS-, GS, CS+ etc.) and how this differed according to trial type. These conditions were modeled using a GLM with the canonical hemodynamic response function (HRF) in SPM12 to examine the association between component time courses and different trial types. The resulting β-weights, a measure of each component's trial-specific amplitude, were entered into statistical analysis to identify those components significantly more engaged in each condition.

**Statistical analysis**. All statistical analysis was carried out in SPSS (Armonk, NY, USA). Levels of conditioning were assessed with paired sample t-tests comparing behavioral risk ratings to CS+ vs. oCS-, and CS+ vs. vCS− in HC and TE separately during pre-acquisition, acquisition and generalization phases. The significance threshold for these behavioral analyses were set at p < 0.0167 to adjust for multiple comparisons using Bonferroni correction (corrected for 3 comparisons).

To examine the generalization phases over time (EG, LG), for both behavior and neural markers, we first assessed changes (delta) in behavioral and neural markers over the two stages (EG-LG) for trauma exposure by comparing TE and HC using a group (TE and HC) by stimulus-type (vCS-, oCS-, GS1, GS2, GS3 and unreinforced CS+) repeated measures ANOVAs, and then assessed the changes of behavior and neural markers over the two stages (EG-LG) of resilience by further comparing TEHC and TEPG with HC using a group (TEPG, TEHC, HC) by stimulus-type (vCS-, oCS-, GS1, GS2, GS3 and unreinforced CS+) repeated measures ANOVAs.

Next, we assessed the changes in steepness of the generalization gradients across early and late stages (i.e., changes/ delta of generalization magnitudes: EG-LG) measured by linear deviation scores (LDS[25]. LDS reflect the degree to which participant level gradients depart from linearity: LDS = ([CS+, CS−] /2) − [GS1, GS2, GS3] /3)), where [CS+, CS−] /2 reflects the theoretical, linear midpoint of the gradient, and [GS1, GS2, GS3] /3 the average response to GSs. This equation provides a single number index reflecting the steepness of generalization gradients, with larger values indicating stronger generalization. We then assessed the behavioral and neural markers of trauma exposure by measuring the delta of LDS across two stages using one-way ANOVA, and then assessed the behavioral and neural markers of resilience by further comparing TEPG and TEHC with HC using one-way ANOVA. Effects of covariate corresponding to different scanners, age and sex was used in all analysis as a covariate of no interest.

For assessing the neural markers of trauma exposure and resilience, the five intrinsic connectivity networks (ICN) meeting selection criteria were used (Table S1). These networks included the salience network (SN), left executive control network (LECN), right executive control network (RECN), anterior default mode network (a-DMN) and posterior default mode network (p-DMN). All neural imaging results were corrected for multiple comparison at $p < 0.01$ (5 networks).

**Statistics and reproducibility**. To ensure the stability of the ICA maps obtained in this study, the infomax algorithm was repeated fifty times with randomly initialized decomposition matrices and the same convergence threshold using the ICASSO approach to assess the reliability of the generated components[47]. For each IC the"centroid" (i.e., the most stable result) was determined following the agglomerative hierarchical clustering with average-linkage criterion, and its consistency was calculated with a cluster quality index ($Iq$) ranging from 0 to 1[47].

**Reporting summary**. Further information on research design is available in the Nature Portfolio Reporting Summary linked to this article.

## Data availability

Data used for visualizations in Figure, 1, 2, 3 are available in Supplementary Data 1. All other datasets generated during the current study are available from the corresponding author on reasonable request.

## Code availability

The code generated during the current study are available from the corresponding author on reasonable request.

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

## Acknowledgements

This research was supported by NIMH grant R01MH105355-01A1 (Dr Neria-PI). Dr. Zhu was supported by K01MH122774 and Brain & Behavior Research Foundation. Dr. Suarez-Jimenez was supported by K01MH118428 and Brain & Behavior Research Foundation.

## Author contributions

X.Z.: Formal behavior and MRI analysis, Writing - original draft; B.S.-J.: Writing - original draft, review & editing; A.L.: Writing - review & editing; S.S.S. and C.M.: Data curation, Project administration; S.S., T.D.W., and M.A.L.: Writing - review & editing; S.L.: Conceptualization, Writing - review & editing, fMRI task supervision; Y.N.: Conceptualization, Funding acquisition, Supervision, Writing - review & editing.

## Competing interests

The authors declare no competing interests.
