## [Peer Review File · Communications Biology]

Reviewers' comments:

Reviewer #1 (Remarks to the Author):

Brief summary of the manuscript: To identify potential behavioral and neural markers of exposure and resilience to trauma, the authors combined fMRI, a generalization task, and a network connectivity approach. In a sample of 114 individuals (88 of whom were analyzed), the authors find behavioral and network connectivity differences between trauma-exposed individuals and healthy controls and network connectivity differences between trauma-exposed individuals with and without psychopathology. The authors note their study suggests potential targets for future treatment of psychopathology.

Overall impression of the work: Overall, I believe this is a strong study that will be a strong contribution to the literature. I can only offer several minor comments to improve this manuscript. My biggest concern is that I am not (yet) convinced these findings are useful for improving treatment rather than improving prevention (which is not mentioned as a potential outlook).

Specific comments:

1) If allowed by the word limit, it could be informative if the authors mentioned the (analyzed) sample size in the abstract. For instance: "Trauma-exposed (TEs; N=...) participants, compared to healthy controls (HCs; N=...) showed lower activity reduction in salience network (SN) and right executive control network (RECN) across the two sequential generalization stages, and worse discrimination learning in SN measured by linear deviation scores (LDS). Comparison of resilient, trauma-exposed healthy participants (TEHCs; N=...), trauma exposed individuals presenting with psychopathology (TEPGs; N=...)"

2) The authors end their abstract with the following implications for future clinical practice: "These findings may indicate a trauma exposure phenotype that has the potential to advance the development of innovative treatments by targeting and engaging specific neural dysfunction among trauma exposed individuals, across different psychopathologies." The implication for practice is also mentioned in the discussion. However, I am not entirely convinced we should consider the findings in this study as a contribution to treatment of individuals with psychopathology. That is, I wonder if the findings in this study suggest (1) targets to reduce existing psychopathology or (2) markers to prevent future psychopathology (e.g., using these markers to select individuals for professions with higher trauma risk such as soldiers). Might it be worth considering what these results mean in the context of prevention?

3) On pg. 4 in the introduction, pattern separation and completion are italic suggesting they are key concepts in the study, but they are only mentioned once. Are they relevant to this specific study at all? If so, I would expect these concepts to return in the discussion. If not, I recommend removing the following sentences from the manuscript altogether and directly move on to the involved brain areas and networks. "Threat generalization and discrimination have been shown to be processed via two distinct computations. The first, pattern separation, takes similar neural activity patterns 86 and converts them to distinct neural representations. The second, pattern completion, operates on these distinct representations and either ignores the difference, if negligible, or generates orthogonal neural representations if the difference is sufficiently large (17)."

4) The authors could consider revising the following sentence by removing computational (which will have a different meaning to different researchers and can be omitted without loss of information) and either (which is a typo if I'm correct). Learning to discriminate is an essential computational process whereby exposure to cues creates either different neural representations for each cue.

5) The following sentence feels slightly off. If the authors refer to "resilient counterparts", they might

also need to mention the non-resilient individuals. "Understanding how these representations are built over time can give us an insight into how psychopathology develops and is maintained and how resilient counterparts overcome it."

6) As mentioned in the supplementary materials, not all participants were analyzed (for various reasons). In the spirit of full transparency, I recommend that the authors mention the number of participants that were analyzed, not only the number of recruited participants.

Reviewer #2 (Remarks to the Author):

The current manuscript is a report from an fMRI study of fear generalization and network connectivity in trauma-exposed individuals with and without psychopathology (e.g., generalized anxiety disorder, PTSD). As compared to healthy controls, trauma-exposed individuals showed less activity reduction in salience and right executive control networks across sequentially presented generalization tasks and impaired discrimination learning in salience network as measured by linear deviation scores. When the authors compared trauma-exposed individuals without psychopathology, trauma-exposed individuals with psychopathology, and healthy controls, they discovered a "resilience" signature of network connectivity differences in the right executive control network during the generalization tasks. This work is compelling and of interest given that the majority of individuals exposed to psychological trauma do not go on to develop psychopathologies such as PTSD.

Specific Comments intended to enhance manuscript prior to publication:

1. Abstract, line 22, suggest revising the end of this line to read, "...by itself results in deficient..."
2. Abstract, line 26, suggest adding the word "putative" between "clarify" and "behavioral"
3. Abstract, line 29, place comma after "(HCs)"
4. Significance Statement, line 44, remove comma after "resilience"
5. Significance Statement, line 45, suggest editing to read, "Findings suggest a trauma signature..."
6. Significance Statement, line 47, suggest editing to read, "...whereas a resilience signature..."
7. Significance Statement, line 50, suggest adding examples of psychopathologies at the end of this sentence.
8. Introduction, line 55 and throughout manuscript, suggest placing a semicolon between parenthetical elements and in-text citation, for example, "(MDD; 1-4)"
9. Introduction, line 58, suggest editing to read, "...across psychopathologies such as those previously listed..."
10. Introduction, line 61, although it is not necessary to identify a specific version of the DSM to make the authors' point, if they do not specify DSM-5 then they should make "Manual" plural to "Manuals"
11. Introduction, line 65, suggest editing to, "...psychopathologies and versus trauma-exposed healthy controls. As such, this focus on PTSD..."
12. Introduction, line 77, suggest providing examples of generalization stimuli that have been used in previous work including concentric circle and faces.
13. Introduction, line 81, please clarify that PTSD is not an anxiety disorder and edit this sentence to include trauma- and stressor-related disorders.
14. Introduction, line 92, suggest adding the word "Functional" before the word "Deficits" starting the second sentence.
15. Introduction, line 108, suggest revising to, "...exposed to trauma with psychiatric illnesses (e.g., PD, GAD, PTSD) or among..."
16. Introduction, line 112, delete the word "either" in this sentence
17. Introduction, line 131, delete comma after "(TE)"
18. Methods, line 158, please provide vendor information (city, state/province, country) for Cedrus
19. Methods, line 159, please provide vendor information (city, state/province, country) for Neurobehavioral Systems
20. Methods, line 173, please provide rationale and reference for selection of an 80% reinforcement

schedule

21. Methods, line 215, please provide vendor information (city, state/province, country) for SPSS (Armonk, NY, USA)

22. Discussion, line 379, revise sentence to, "...does not necessarily hamper one's ability..."

23. Discussion, line 383, delete the word "showed" from this sentence

24. Discussion, at the close of this section it is not clear if the authors' suggestion that one could induce neurogenesis specific to salience and executive control networks and suggestion of beneficial use of tDCS are independent thoughts or if they are suggesting that tDCS could induce neurogenesis?

Reviewer #3 (Remarks to the Author):

In this manuscript, the effects of exposure to trauma are investigated in trauma-exposed patients and the differences between the patients who developed psychopathology and resilient are examined using both behavioral and neuroimaging (fMRI) measures. Authors found signatures of trauma exposure in salience network (SN) and right executive control network (RECN) in patients, and they found differences in these networks for resilient when compared with trauma-exposed individuals presenting with psychopathology. While the goal of the paper in differentiating the two population of trauma-exposed individuals is interesting and definitely raises potentially important application especially when treating the patients of this kind, there are issues surrounding the manuscript organization which compromises the impact in its current form. The most important issues are the exploratory nature of it rather than having a specific hypothesis and largely vague interpretation of results and explanation of tasks and methods. Also, despite the fact that their dataset includes a range of patients exposed to trauma instead of focusing on a specific psychopathology, I believe the results they've found are largely reported previously in the literature (A Kraus-Utz, et al 2014; Van der werf et al., 2013; Chen et al., 2013; Korgaonkar et al., 2020). Therefore, I believe at least the following important gaps should be resolved before publishing the current form.

Major comments:

1. The task is unclear. The experiment consists of many trials incorporated in many conditions. However, it is explained in a vague way without any appropriate visualization of it. I believe adding a new figure (or maybe modifying figure S6) which simultaneously shows one trial with timing along with the overall design can be very helpful. Right now, the writing is unclear, and it is very hard to follow the sequence of events. The figure will hopefully highlight the difference between six conditions used in the experiment and help the readers to differentiate between different conditions and consequently between different analysis.
2. Wrong reference to figure/tables. Many places in the manuscript, authors refer to wrong figure/tables. For example, page 6, line 140 or line 142 (figure 4S is used, but it should be figure 5S). Authors should carefully check for the figure/table references in the write-up and ensure that it appears correctly everywhere in the manuscript.
3. Unclear explanation of neural activity changes results. On page 12, line 285, authors indicate that "This main effect of group was driven by HC displaying a higher reduction of SN and RECN activity, compared to TE who displayed consistently higher SN and RCEN activity, over the two generalization phases (Figure 2)." From merely looking at Figure 2, it is not clear why the authors claim that TE showed higher SN and RECN activity. Can the authors please clarify on this? A similar claim is made in page 13, line 325. Since figure 2 only shows the EG-LG contrast, this type of interpreting the results is misleading. Therefore, authors should either provide another figure showing the activations in each stage separately, or simply change the interpretation of their results.

Minor concerns

1. Page 8, line 182. The TR is relatively small for functional data acquisition. Since the total number of slices are 27, I speculate that the data acquisition was not from the whole brain. Can authors confirm that the data acquisition was from whole brain? If not, why?

2. Page 11, line 266. It's better to report t/F values from t-test/ANOVA.
3. What does the error bars indicate in figures? Standard error of mean?
4. A thorough explanation in caption of figure S5 is needed. Right now, the flow in the chart is not straightforward and reader needs to go through the whole methods section to figure out what exactly figure S5 is showing. However, I found this figure very helpful and I believe with a better caption, it'll be very informative. Overall, figures S1-5 would be better if the captions reflect the purpose of them in a more informative way.
5. Page 15, line 375. where  were

Reviewer #1 (Remarks to the Author):

Brief summary of the manuscript: To identify potential behavioral and neural markers of exposure and resilience to trauma, the authors combined fMRI, a generalization task, and a network connectivity approach. In a sample of 114 individuals (88 of whom were analyzed), the authors find behavioral and network connectivity differences between trauma-exposed individuals and healthy controls and network connectivity differences between trauma-exposed individuals with and without psychopathology. The authors note their study suggests potential targets for future treatment of psychopathology.

Overall impression of the work: Overall, I believe this is a strong study that will be a strong contribution to the literature. I can only offer several minor comments to improve this manuscript. My biggest concern is that I am not (yet) convinced these findings are useful for improving treatment rather than improving prevention (which is not mentioned as a potential outlook).

[Response]: We appreciate the Reviewer's thorough review. We have now included the potential impact of this study on prevention to the discussion (see our response to comment 2 below).

COMMENT 1

If allowed by the word limit, it could be informative if the authors mentioned the (analyzed) sample size in the abstract. For instance: "Trauma-exposed (TEs; N=...) participants, compared to healthy controls (HCs; N=...) showed lower activity reduction in salience network (SN) and right executive control network (RECN) across the two sequential generalization stages, and worse discrimination learning in SN measured by linear deviation scores (LDS). Comparison of resilient, trauma-exposed healthy participants (TEHCs; N=...), trauma exposed individuals presenting with psychopathology (TEPGs; N=...)"

RESPONSE 1

Thank you for this suggestion, we have added the sample size in the analysis in the abstract as follows: *"Trauma-exposed (TEs; N=62) participants, compared to healthy controls (HCs; N=26), showed lower activity reduction in salience network (SN) and right executive control network (RECN) across the two sequential generalization stages, and worse discrimination learning in SN measured by linear deviation scores (LDS). Comparison of resilient, trauma-exposed healthy participants (TEHCs; N=31), trauma exposed individuals presenting with psychopathology (TEPGs; N=31), and HCs, revealed a resilience signature of network connectivity differences in the RECN during generalization learning measured by LDS."*

COMMENT 2

The authors end their abstract with the following implications for future clinical practice: "These findings may indicate a trauma exposure phenotype that has the potential to advance the development of innovative treatments by targeting and engaging specific neural dysfunction among trauma exposed individuals, across different psychopathologies." The implication for practice is also mentioned in the discussion. However, I am not entirely convinced we should consider the findings in this study as a contribution to treatment of individuals with psychopathology. **That is, I wonder if the findings in this study suggest (1) targets to reduce existing psychopathology or (2) markers to prevent future psychopathology (e.g., using these markers to select individuals for professions with higher trauma risk such as soldiers). Might it be worth considering what these results mean in the context of prevention?**

RESPONSE 2

We thank the reviewer for this thought-provoking comment. We agree that it is interesting to think about how these results might be implicated in prevention or the identification of potential risk-factors that could lead to psychopathology. However, since this study is cross-sectional in design, we cannot guarantee whether these brain patterns would be observed prior to trauma or the development of psychopathology. Nonetheless, we have added this as a limitation and potential avenue for future research in the Discussion page 18, line 493, as follows: *"Finally, and most interesting, it is tempting to consider the current results as potential targets for prevention or identifying risk-factors for the development of psychopathology after trauma-exposure. However, due to the limitations of cross-sectional studies, such as this one, we cannot infer whether the observed results in this study are caused by trauma-exposure or are developed prior to trauma-exposure. Future longitudinal studies should focus on these neural signatures, to identify their potential applicability for prevention of future psychopathologies."*

COMMENT 3

On page 4 in the introduction, pattern separation and completion are italic suggesting they are key concepts in the study, but they are only mentioned once. Are they relevant to this specific study at all? If so, I would expect these concepts to return in the discussion. If not, I recommend removing the following sentences from the manuscript altogether and directly move on to the involved brain areas and networks. "Threat generalization and discrimination have been shown to be processed via two distinct computations. The first, pattern separation, takes similar neural activity patterns and converts them to distinct neural representations. The second, pattern completion, operates on these distinct representations and either ignores the difference, if negligible, or generates orthogonal neural representations if the difference is sufficiently large (17)."

RESPONSE 3

Pattern separation and completion are two processes that aid the overall process of threat discrimination. However, noting the reviewer's comment, that indeed we do not refer to these processes again we have opted to remove the following sentences, without loss of important information: *"Threat generalization and discrimination have been shown to be processed via two distinct computations. The first, pattern separation, takes similar neural activity patterns and converts them to distinct neural representations. The second, pattern completion, operates on these distinct representations and either ignores the difference, if negligible, or generates orthogonal neural representations if the difference is sufficiently large (17)."*

The paragraph now reads as follows: *“Several brain areas have been identified to be involved in distinct computational processes that are responsible for threat generalization and discrimination. These include the hippocampus (18, 19), ventral prefrontal cortex (vPFC; 1, 13), insula (20), dorsomedial prefrontal cortex (dmPFC; 21), anterior cingulate cortex (ACC), and thalamus (20). Functional deficits in these brain areas, related to generalization/discrimination, have been attributed to PTSD (22, 23). The involvement of these neural areas also underscores the potential role of several essential brain networks in generalization/discrimination, including the default mode network (DMN), salience network (SN), and executive control network (ECN). These brain networks were also shown to play a key role in memory (DMN; 24), bottom-up (SN) and top-down executive control processes (ECN 25).”*

COMMENT 4

The authors could consider revising the following sentence by removing computational (which will have a different meaning to different researchers and can be omitted without loss of information) and either (which is a typo if I'm correct). Learning to discriminate is an essential computational process whereby exposure to cues creates either different neural representations for each cue.

RESPONSE 4

Thank you for this suggestion, we have revised the sentence (page 5) and now reads as follows: *“Learning to discriminate is an essential process whereby exposure to cues creates different neural representations for each cue.”*

COMMENT 5

The following sentence feels slightly off. If the authors refer to "resilient counterparts", they might also need to mention the non-resilient individuals. "Understanding how these representations are built over time can give us an insight into how psychopathology develops and is maintained and how resilient counterparts overcome it."

RESPONSE 5

We referred the psychopathology group as the resilient counterparts. We have now revised the sentence as follows: *“Understanding how these representations are built over time can give us an insight into how psychopathology develops and is maintained and how resilient trauma-exposed people overcome it.”*

COMMENT 6

As mentioned in the supplementary materials, not all participants were analyzed (for various reasons). In the spirit of full transparency, I recommend that the authors mention the number of participants that were analyzed, not only the number of recruited participants.

RESPONSE 6

To improve the transparency in the main manuscript, we have moved the exclusion section from the supplementary material to the Methods in the Participants section Page 6, as follows: *“From the original 114 sample, four participants were excluded due to being late for scan and not able to finish the SG task (two participants), or task failure of delivering shock (two participants). Twenty-two additional participants (15 TE and 7 HC) were excluded with greater risk rating to vCS-, compared to CS+ during acquisition phase in the fMRI task, or greater rating to vCS-, compared to red cross in post questionnaires. No participant was excluded from further analysis because of movement exceeding ± 1 mm. Consequently, the final neural imaging analysis included 88 participants: 62 TE and 26 HC. Sum of root mean square (RMS) of 6 relative head motion parameters (movement from this time point to the*

next one) was calculated for each participant in all groups (TEPG, TEHC, and HC). No significant difference in head motion was found between each pair of groups ($p>0.5$). Linear regression was performed to study the linear relationship between the dependent variable (sum of head motion) and independent variable (groups). The regression analysis results showed that the total head motion could not predict groups ($p>0.05$)."

We have now listed only the number of participants in the final analysis in the abstract, as follows: "Trauma-exposed (TEs; N=62) participants, compared to healthy controls (HCs; N=26), showed lower activity reduction in salience network (SN) and right executive control network (RECN) across the two sequential generalization stages, and worse discrimination learning in SN measured by linear deviation scores (LDS). Comparison of resilient, trauma-exposed healthy participants (TEHCs; N=31), trauma exposed individuals presenting with psychopathology (TEPGs; N=31), and HCs, revealed a resilience signature of network connectivity differences in the RECN during generalization learning measured by LDS."

We create a new demographic table 1 for these participants in the final analysis (N=88 total), as follows:

		HC	TE	TEPG	TEHC
N		26	62	31	31
Sex, N	Male	15	37	19	18
	Female	11	25	12	13
Race, N	Black/African American	12	29	13	16
	White/Caucasian	10	15	9	6
	Asian	1	6	2	4
	Others	2	12	7	5
Age, mean years (SD)		34.58 (11.56)	37.05 (12.35)	37.19 (13.06)	36.9 (11.81)
Education, mean (SD)		15.83 (2.64)	14.65 (2.47)	14.36 (2.78)	14.93 (2.14)
CAPS, mean (SD)		-	17.74 (16.36)	31.03 (11.95)	4.45 (6.03)
HAM-D, mean (SD)		0.4 (0.82)	7.16 (7.26)	12.43 (6.78)	2.06 (2.56)
LSAS, mean (SD)		11.54 (13.29)	40.05 (35.06)	59.55 (34.11)	19.90 (22.66)
PDSS, mean (SD)		0.46 (1.84)	5.2 (6.16)	8.97 (6.18)	1.30 (2.84)
SF36, mean (SD)		87.85 (7.63)	66.62 (23.39)	51.23 (20.28)	82.53 (13.84)
GAD-7, mean (SD)		1.08 (2.43)	6.15 (6.16)	9.94 (5.76)	2.23 (3.60)
Current	PTSD	0	26	26	0
	MDD	0	11	11	0
	PDD	0	5	5	0
	SAD	0	1	1	0
	PD	0	3	3	0
	OCD	0	1	1	0
	ADHD	0	1	1	0
	Insomnia	0	1	1	0

Eating Disorder	0	1	1	0
Subthreshold PTSD	0	2	2	0
Subthreshold MDD	0	2	2	0

Reviewer #2 (Remarks to the Author):

The current manuscript is a report from an fMRI study of fear generalization and network connectivity in trauma-exposed individuals with and without psychopathology (e.g., generalized anxiety disorder, PTSD). As compared to healthy controls, trauma-exposed individuals showed less activity reduction in salience and right executive control networks across sequentially presented generalization tasks and impaired discrimination learning in salience network as measured by linear deviation scores. When the authors compared trauma-exposed individuals without psychopathology, trauma-exposed individuals with psychopathology, and healthy controls, they discovered a “resilience” signature of network connectivity differences in the right executive control network during the generalization tasks. This work is compelling and of interest given that the majority of individuals exposed to psychological trauma do not go on to develop psychopathologies such as PTSD.

Specific Comments intended to enhance manuscript prior to publication:

[Response]: We thank the Reviewer’s thorough review. We have edited the manuscripts based on the suggestions made. See below for the point-by-point edits.

COMMENT 1

Abstract, line 22, suggest revising the end of this line to read, “...by itself results in deficient...”

RESPONSE 1

We have incorporated the suggestion and edited the Abstract, line 22, to read as follows: *“While impaired fear generalization is known to underlie a wide range of psychopathology, the extent to which exposure to trauma by itself results in deficient fear generalization and its neural abnormalities is yet to be studied.”*

COMMENT 2

Abstract, line 26, suggest adding the word “putative” between “clarify” and “behavioral”

RESPONSE 2

We have incorporated the suggestion and edited the Abstract, line 26, to read as follows: *“Here, we utilized fMRI, a generalization task, and a network connectivity approach to clarify putative behavioral and neural markers of trauma and resilience.”*

COMMENT 3

Abstract, line 29, place comma after “(HCs)”

RESPONSE 3

We have incorporated the suggestion and edited the Abstract, line 29, to read as follows: *“Trauma-exposed (TEs; N=62) participants, compared to healthy controls (HCs; N=26), showed lower activity reduction in salience network (SN) and right executive control network (RECN) across the two sequential*

generalization stages, and worse discrimination learning in SN measured by linear deviation scores (LDS)."

COMMENT 4

Significance Statement, line 44, remove comma after "resilience"

RESPONSE 4

We have incorporated the suggestion and edited the Significance Statement, line 44, to read as follows: *"The present study aimed to identify unique behavioral and neural markers of trauma and resilience by utilizing fMRI and a generalization/discrimination task."*

COMMENT 5

Significance Statement, line 45, suggest editing to read, "Findings suggest a trauma signature..."

RESPONSE 5

We have incorporated the suggestion and edited the Significance Statement, line 45, to read as follows: *"Findings suggest a trauma signature may involve enhanced threat detection and monitoring network engagement via the salience network (SN), whereas a resilience signature engages the right executive network (RECN), which may serve as a compensatory mechanism."*

COMMENT 6

Significance Statement, line 47, suggest editing to read, "...whereas a resilience signature..."

RESPONSE 6

We have incorporated the suggestion and edited the Significance Statement, line 47, to read as follows: *"whereas a resilience signature engages the right executive network (RECN), which may serve as a compensatory mechanism."*

COMMENT 7

Significance Statement, line 50, suggest adding examples of psychopathologies at the end of this sentence.

RESPONSE 7

We have incorporated the suggestion and edited the Significance Statement, line 51, to read as follows: *"Findings have the potential to advance novel treatment development by targeting neural dysfunction among trauma-exposed individuals, across different psychopathologies, such as PTSD."*

COMMENT 8

Introduction, line 55 and throughout manuscript, suggest placing a semicolon between parenthetical elements and in-text citation, for example, "(MDD; 1-4)"

RESPONSE 8

We have incorporated the suggestion and edited the Introduction, line 55 and throughout manuscript, to read as follows: *"including posttraumatic stress disorder (PTSD), panic disorder (PD), generalized anxiety disorder (GAD), and major depression disorder (MDD; 1-4)."*

COMMENT 9

Introduction, line 58, suggest editing to read, "...across psychopathologies such as those previously listed..."

RESPONSE 9

We have incorporated the suggestion and edited the Introduction, line 58, to read as follows: *"Addressing the question of whether trauma exposure is associated with a distinct neural signature across psychopathologies such as those previously listed may advance our understanding of the corresponding neural dysfunction."*

COMMENT 10

Introduction, line 61, although it is not necessary to identify a specific version of the DSM to make the authors' point, if they do not specify DSM-5 then they should make "Manual" plural to "Manuals"

RESPONSE 10

We have incorporated the suggestion and edited the Introduction, line 64, to read as follows: *"such as the Diagnostic and Statistical Manuals of Mental Disorders (DSM; 8)."*

COMMENT 11

Introduction, line 65, suggest editing to, "...psychopathologies and versus trauma-exposed healthy controls. As such, this focus on PTSD..."

RESPONSE 11

We have incorporated the suggestion and edited the Introduction, line 68, to read as follows: *"limiting our understanding of potential shared mechanisms across psychopathologies versus trauma-exposed healthy controls. As such, this focus on PTSD has also complicated efforts to clarify whether resilience (i.e., exposure to trauma without developing significant psychiatric symptoms) may have a unique neural signature, with research on resilience being relatively sparse (10-12)."*

COMMENT 12

Introduction, line 77, suggest providing examples of generalization stimuli that have been used in previous work including concentric circle and faces.

RESPONSE 12

We have incorporated the suggestion to provide an example and edited the Introduction, line 86, to read as follows: *"Generalization transfers a conditioned response from one stimulus to another similar, yet different, stimulus, such as concentric circles (15)."*

COMMENT 13

Introduction, line 81, please clarify that PTSD is not an anxiety disorder and edit this sentence to include trauma- and stressor-related disorders.

RESPONSE 13

We have incorporated the suggestion to include trauma and stress related disorders and edited the Introduction, line 91, to read as follows: *"A recent meta-analysis study has shown significantly heightened behavioral fear generalization across trauma-, stress-, and anxiety-related disorder participants including GAD, PTSD, PD, SAD than controls, however, this study was limited to behavioral and psychophysiological data only (16)."*

COMMENT 14

Introduction, line 92, suggest adding the word “Functional” before the word “Deficits” starting the second sentence.

RESPONSE 14

We have incorporated the suggestion and edited the Introduction, line 98, to read as follows: *“Functional deficits in these brain areas, related to generalization/discrimination, have been attributed to PTSD (22, 23).”*

COMMENT 15

Introduction, line 108, suggest revising to, “...exposed to trauma with psychiatric illnesses (e.g., PD, GAD, PTSD) or among...”

RESPONSE 15

We have incorporated the suggestion and edited the Introduction, line 138, to read as follows: *“However, no study to date has assessed this neural marker among people exposed to trauma with psychiatric illnesses or among resilient participants who were exposed to trauma.”*

COMMENT 16

Introduction, line 112, delete the word “either” in this sentence

RESPONSE 16

We have incorporated the suggestion and edited the Introduction, line 143, to read as follows: *“Learning to discriminate is an essential process whereby exposure to cues creates different neural representations for each cue.”*

COMMENT 17

Introduction, line 131, delete comma after “(TE)”

RESPONSE 17

We have incorporated the suggestion and edited the Introduction, line 144, to read as follows: *“Understanding how these representations are built over time can give us an insight into how psychopathology develops and is maintained and how resilient trauma-exposed counterparts overcome it.”*

COMMENT 18

Methods, line 158, please provide vendor information (city, state/province, country) for Cedrus

RESPONSE 18

We have incorporated the suggestion and edited the Methods, line 207, to read as follows: *“Lumina LP-404 by Cedrus, San Pedro, Los Angeles, USA”*

COMMENT 19

Methods, line 159, please provide vendor information (city, state/province, country) for Neurobehavioral Systems

RESPONSE 19

We have incorporated the suggestion and edited the Methods, line 209, to read as follows:
"Neurobehavioral Systems, Berkeley, CA, USA"

COMMENT 20

Methods, line 173, please provide rationale and reference for selection of an 80% reinforcement schedule

RESPONSE 20

We have now added reference for selection of an 80% reinforcement schedule and edited the Methods, line 226, to read as follows: *"(80% reinforcement schedule; 14)"*

COMMENT 21

Methods, line 215, please provide vendor information (city, state/province, country) for SPSS (Armonk, NY, USA)

RESPONSE 21

We have incorporated the suggestion and edited the Methods, line 270, to read as follows: *"All statistical analysis was carried out in SPSS (Armonk, NY, USA)."*

COMMENT 22

Discussion, line 379, revise sentence to, *"...does not necessarily hamper one's ability..."*

RESPONSE 22

We have incorporated the suggestion and edited the Discussion, line 438, to read as follows: *"These results suggest that trauma exposure by itself does not necessarily hamper one's ability to discriminate between cues, unless one develops psychopathology following exposure to trauma."*

COMMENT 23

Discussion, line 383, delete the word "showed" from this sentence

RESPONSE 23

We have incorporated the suggestion and edited the Discussion, line 442, to read as follows: *"Studying averaged generalization across the two stages in TE participants maintained higher overall activity in the SN, indicating a general trauma exposure marker."*

COMMENT 24

Discussion, at the close of this section it is not clear if the authors' suggestion that one could induce neurogenesis specific to salience and executive control networks and suggestion of beneficial use of tDCS are independent thoughts or if they are suggesting that tDCS could induce neurogenesis?

RESPONSE 24

We have clarified that the mechanisms of tDCS in the Discussion, line 551, as follows: *"Emerging research suggests that brain stimulation, particularly transcranial direct current stimulation (tDCS), a non-invasive technique, which elicited its action via multifaceted mechanisms, such as immunomodulation and neurogenesis, is a promising treatment for fear-related abnormalities by modulating threat memories and enhancing cognitive control via application to the prefrontal cortex."*

Reviewer #3 (Remarks to the Author):

In this manuscript, the effects of exposure to trauma are investigated in trauma-exposed patients and the differences between the patients who developed psychopathology and resilient are examined using both behavioral and neuroimaging (fMRI) measures. Authors found signatures of trauma exposure in salience network (SN) and right executive control network (RECN) in patients, and they found differences in these networks for resilient when compared with trauma-exposed individuals presenting with psychopathology. While the goal of the paper in differentiating the two population of trauma-exposed individuals is interesting and definitely raises potentially important application especially when treating the patients of this kind, there are issues surrounding the manuscript organization which compromises the impact in its current form. **The most important issues are the exploratory nature of it rather than having a specific hypothesis and largely vague interpretation of results and explanation of tasks and methods.** Also, despite the fact that their dataset includes a range of patients exposed to trauma instead of focusing on a specific psychopathology, **I believe the results they've found are largely reported previously in the literature** (A Kraus-Utz, et al 2014; Van der werf et al., 2013; Chen et al., 2013; Korgaonkar et al., 2020). Therefore, I believe at least the following important gaps should be resolved before publishing the current form.

[Response]: We have now edited the introduction and included the hypothesis on page 6 as follows:
“For behavioral markers, we hypothesized that there would be elevated generalization to stimuli resembling the CS+ in the TEPG group, compared to HC or TEHC groups. For neural markers, we expected that in the TEPG group there would be limited changes of network connectivity in SN and ECN over the two generalization stages, compared to TEHC or HC group.” We also state the gaps of knowledge as follows: “However, no study to date has assessed this neural marker among people exposed to trauma with psychiatric illnesses or among resilient participants who were exposed to trauma, but did not develop any significant psychiatric symptoms. In addition, no study to date has looked at how trauma exposed individuals learn to discriminate between cues over time.”

We appreciate the reviewer suggestion on the literature (A Kraus-Utz, et al 2014; Van der werf et al., 2013; Chen et al., 2013; Korgaonkar et al., 2020), however, we could not locate the specific and relevant papers from those authors. In our aim to increase clarity and completeness of the paper, would the reviewer be able to provide the titles of the papers from these authors?

MAJOR COMMENT 1

The task is unclear. The experiment consists of many trials incorporated in many conditions. However, it is explained in a vague way without any appropriate visualization of it. I believe adding a new figure (or maybe modifying figure S6) which simultaneously shows one trial with timing along with the overall design can be very helpful. Right now, the writing is unclear, and it is very hard to follow the sequence of events. The figure will hopefully highlight the difference between six conditions used in the experiment and help the readers to differentiate between different conditions and consequently between different analysis.

RESPONSE

We have now included a figure in S6 which shows one trial with timing along with the overall design for each phase.

Figure S6: A. This task consists of six types of stimuli including conditioned stimuli (CS+: conditioned danger cue; oCS-: o shape conditioned safe cue; vCS-: v shape conditioned safe cue) and three generalization stimuli (GSs including GS1, GS2, GS3). Half of the participants were assigned to counterbalancing group A and half to B. For both counterbalancing groups A and B, GS3 consisted of the ring closest in size to CS., with GS2 and GS1 further decreasing in similarity to CS+. B. An example event sequence for CS+. There were a string of colored crosshairs (blue, yellow, red, green, and purple) presented serially for a duration of 800 ms each, in a quasi-random order in the center of the viewing screen during 4 s presentation of each stimulus. The inter-trial interval (ITI) periods last 2.4 s (three crosshairs) or 4.8 s (four crosshairs). Participants were instructed to continuously monitor the stream of colored crosshairs and rate their perceived level of risk for shock as quickly as possible following each red cross using a three-button fiber optic response pad, where 0='no risk', 1='moderate risk' and 2='high risk'. C. The generalization paradigm timeline included three phases; (i) pre-acquisition-consisting in the absence of any shock US; (ii) acquisition-including with shock at an 80% reinforcement schedule; and (iii) generalization-including an early generalization (EG) stage and late generalization (LG) stage with shock US at a 33% reinforcement schedule to prevent extinction of the conditioned response.

MAJOR COMMENT 2

Wrong reference to figure/tables. Many places in the manuscript, authors refer to wrong figure/tables. For example, page 6, line 140 or line 142 (figure 4S is used, but it should be figure 5S). Authors should carefully check for the figure/table references in the write-up and ensure that it appears correctly everywhere in the manuscript.

RESPONSE

We thank the Reviewer for pointing out this oversight. We have now edited the order of the figures/table references throughout the manuscript.

MAJOR COMMENT 3

Unclear explanation of neural activity changes results. On page 12, line 285, authors indicate that “This main effect of group was driven by HC displaying a **higher reduction** of SN and RECN activity, compared to TE who displayed consistently higher SN and RCEN activity, over the two generalization phases (Figure 2).” From merely looking at Figure 2, it is not clear why the authors claim that TE showed higher SN and RECN activity. Can the authors please clarify on this? A similar claim is made in page 13, line 325. Since figure 2 only shows the EG-LG contrast, this type of interpreting the results is misleading. Therefore, authors should either provide another figure showing the activations in each stage separately, or simply change the interpretation of their results.

RESPONSE

We thank the Reviewer for this suggestion. To better quantify the changes over the two generalization phases, we used the delta measures, in which we assessed the changes of neural activity over the two stages (delta: EG-LG). Therefore, we reported “This main effect of group was driven by HC displaying a higher reduction of SN and RECN activity”, as shown in Figure 2, we presented the delta changes for each network, blue line for HC and red line for TE.

To increase clarity in this very important point, we have now improved the figure captions as follows:
“Figure 2: Neural activity changes (delta: EG-LG) over the two generalization phases to conditioned and generalization stimuli (vCS-, oCS-, GS1, GS2, GS3, CS+) for each network across two stages.”

Additionally, we provided another figure showing the activations in each stage separately (Figure S2):

Figure S2: Neural responses to conditioned and generalization stimuli (vCS-, oCS-, GS1, GS2, GS3, CS+) across ICNs by stages across HC and TE groups.

Finally, we have also referred the Figure S2 in the results section on page 12 as follows: *“This main effect of group was driven by HC displaying a higher reduction of SN and RECn activity, compared to TE who remained consistently higher SN and RECn activity, over the two generalization phases (Figure 2, Figure S2).”*

MINOR COMMENT 1

Page 8, line 182. The TR is relatively small for functional data acquisition. Since the total number of slices are 27, I speculate that the data acquisition was not from the whole brain. Can authors confirm that the data acquisition was from whole brain? If not, why?

RESPONSE 1

We have now clarified how the whole brain T2*-weighted echo-planar images (EPIs) were collected. We selected a shorter TR, to better measure the dynamic changes of functional connectivity. We have edited Page 8, line 182, as follows: *“Whole brain T2*-weighted echo-planar images (EPIs) depicting the blood-oxygen-level-dependent (BOLD) were acquired for each participant.”*

MINOR COMMENT 2

Page 11, line 266. It's better to report t/F values from t-test/ANOVA.

RESPONSE 2

We have now reported t/F values in the results section Page 11-12, line 320, as follows: *“Results showed no CS+ vs. vCS- differences at the pre-acquisition phase in both groups (HC: $p=0.09$, $t=1.75$; TE: $p=0.39$, $t=-0.88$). Conversely, a significant difference between the cues at acquisition (HC: $p=0.009$, $t=-2.84$; TE: $p<0.001$, $t=-8.01$) and generalization phase (HC: $p=0.001$, $t=-3.83$; TE: $p<0.001$, $t=-9.95$) emerged in both groups (Figure S1).”*

MINOR COMMENT 3

What does the error bars indicate in figures? Standard error of mean?

RESPONSE 3

The error bars stand for +/- 2 standard error (SE), we have added this information in all figure captions as follows: *“Figure 1: Risk rating in early SG phase (left), late SG phase (middle), and delta changes over time (EG-LG), error bar stand for +/- 2 standard error (SE)”*; *“Figure 2: Neural activity changes (delta: EG-LG) over the two generalization phases to conditioned and generalization stimuli (vCS-, oCS-, GS1, GS2, GS3, CS+) for each network across two stages. error bar stand for +/- 2 standard error (SE).”*; *“Figure 3: LDS changes in different groups for SN and RECN. error bar stand for +/- 2 standard error (SE).”*

MINOR COMMENT 4

A thorough explanation in caption of figure S5 is needed. Right now, the flow in the chart is not straightforward and reader needs to go through the whole methods section to figure out what exactly figure S5 is showing. However, I found this figure very helpful and I believe with a better caption, it'll be very informative. Overall, figures S1-5 would be better if the captions reflect the purpose of them in a more informative way.

RESPONSE 4

We thank the Reviewer for the suggestion, we have expanded on the caption for figure S1-5.

Figure S1 reads as follows: *“Figure S1: Mean behavioral risk ratings for pre-ACQ phase, ACQ phase and SG phase for two group (top: HCs, TEs) and three groups (bottom: HCs, TEHCs, TEPGs). Results suggested all groups learned the CS contingencies.”*

”

Figure S2 reads as follows: *“Figure S2: Neural responses to conditioned and generalization stimuli (vCS-, oCS-, GS1, GS2, GS3, CS+) across ICNs by two stages across HC and TE groups.*

Abbreviation: CS+: conditioned stimuli, danger cue; oCS-: conditioned stimuli, o shape safe cue; vCS-: conditioned stimuli, v shape safe cue; GS: generalization stimuli; SN: salience network; RECN: right executive control network; LECN: left executive control network; A-DMN: anterior default mode network; p-DMN: posterior default mode network; HC: non-trauma-exposed healthy controls ; TEHC: trauma-exposed healthy controls; TEPG: trauma-exposed psychopathology group.”

Figure S3 reads as follows: *“Figure S3: Neural responses to conditioned and generalization stimuli (vCS-, oCS-, GS1, GS2, GS3, CS+) across ICNs by two stages across HC and TEHC and TEPG groups.*

Abbreviation: CS+: conditioned stimuli, danger cue; oCS-: conditioned stimuli, o shape safe cue; vCS-: conditioned stimuli, v shape safe cue; GS: generalization stimuli; SN: salience network; RECN: right executive control network; LECN: left executive control network; A-DMN: anterior default mode network; p-DMN: posterior default mode network; HC: non-trauma-exposed healthy controls ; TEHC: trauma-exposed healthy controls; TEPG: trauma-exposed psychopathology group.”

Figure S4 reads as follows: *“Figure S4: LDS across two stages in different groups for SN and RECN. We assessed the changes in steepness of the generalization gradients across early and late stages (i.e., changes/ delta of generalization magnitudes: EG-LG) measured by linear deviation scores (LDS). SN: Salience network, RECN: right executive control network.”*

Figure S5 reads as follows: *“Figure S5: Flowchart of study protocol. 668 participants were screened for this protocol, among these, 483 participants did not meet inclusion criteria. 185 participants were assessed for eligibility and 123 participants were enrolled in this study. 114 participants (79 TEs and 35 HCs) completed both baseline clinical assessment and the fMRI assessment. Of the 79 TE participants, 41 had psychopathology at the level of a psychiatric diagnosis (TEPG), and 38 were trauma-exposed healthy controls with no diagnosis (TEHC). From the 114 sample, four participants were excluded due to being late for scan and not able to finish the SG task (two participants), or task failure of delivering shock (two participants). Twenty-two additional participants (15 TE and 7 HC) were excluded with greater risk rating to vCS-, compared to CS+ during acquisition phase in the fMRI task, or greater risk rating to vCS-, compared to red cross in post questionnaires. Consequently, the final neural imaging analysis included 88 participants: 62 TE and 26 HC.”*

MINOR COMMENT 5

Page 15, line 375. where  were

RESPONSE 5

We have incorporated the suggestion and edited Page 15, line 375, to read as follows: *“In turn, resilient trauma-exposed counterparts were able to discriminate between the cues comparably to HCs.”*

REVIEWERS' COMMENTS:

Reviewer #1 (Remarks to the Author):

I am satisfied with how the authors addressed my suggestions. I have no further comments.

Reviewer #2 (Remarks to the Author):

The authors have made substantial changes to the manuscript that were highly responsive to the Reviewers' comments. No more concerns remain from this Reviewer.

Reviewer #3 (Remarks to the Author):

The authors have satisfactorily addressed the points raised in my original review. I now recommend the paper for publication at Communications Biology.